# Exploring Factors Influencing Recreational Experiences of Urban River Corridors Based on Social Media Data

Lin Shi [1], Sreetheran Maruthaveeran [1,*], Mohd Johari Mohd Yusof [1] and Chenyang Dai [1,2]

1    Department of Landscape Architecture, Faculty of Design & Architecture, Universiti Putra Malaysia, Serdang 43400, Selangor Darul Ehsan, Malaysia
2    Department of Design, Faculty of Arts, Hebei University of Economics & Business, Shijiazhuang 050061, China
*    Correspondence: sreetheran@upm.edu.my; Tel.: +60-12-3310942

**Abstract:** River corridors, recognized as "blue–green infrastructure," have become a crucial support system for urban sustainability in contemporary urbanized societies. Understanding the factors influencing the recreational experience along urban river corridors is paramount for enhancing visitors' health and well-being. This study focuses on the Hutuo River Corridor in Shijiazhuang, China, collecting 3006 valid reviews from Dianping, a prominent review platform. We developed a text-based thematic model and conducted content analysis using this dataset. The main social (visiting time, duration of stay, motivation, safety, and visitors' types and activities) and physical (natural elements, artificial facilities, maintenance and management, accessibility, distance, models of transportation, weather, and seasons) factors associated with recreational experiences were identified. We assessed visitor perceptions of urban river corridors and elucidated facilitators or barriers through textual content analysis in reviews. The results indicate the feasibility of employing social media data to study visitors' recreational experiences along urban river corridors. This comprehensive exploration from a qualitative ecological perspective contributes valuable insights for urban planning and management. Moreover, the findings hold significant implications for understanding the usage patterns of river corridors in China and potentially in other countries.

**Keywords:** urban river corridor; leisure experiences; recreational experiences; socio-ecological model; social media data

## 1. Introduction

In modern urbanized society, river corridors have become essential support systems for the sustainable development of cities [1]. Their significance is heightened as most of the global population converges in densely populated areas. City dwellers, profoundly influenced by rapid demographic shifts and the pressures associated with urban lifestyles, actively seek ways to reconnect with nature [2,3]. Creating diverse public spaces along riverbanks has led to a surge in recreational activities, fostering intricate interactions with riverine ecosystems.

River corridors, known as "blue–green infrastructure", are vital to the stability, comfort, and sustainability of urban areas [4,5]. Firstly, functioning as natural systems within cities, they regulate the urban microclimate, mitigate the urban heat island effect, maintain water and soil, purify water bodies, and enhance biodiversity by creating buffer zones of riverbank vegetation [6–12]. Secondly, the recreational development of rivers in most urban settings follows a "ribbon" pattern, with rivers constituting a fundamental element of the urban dynamic [13]. Geographically, they dominate urban regions, delineating the framework and boundaries of cities, serving as cultural heritage and urban landmarks [14]. The river corridor landscape, by collecting water sources, creating visual transparency in architectural spaces, and increasing aesthetic value, enhances the quality of urban spaces [15,16]. Additionally, urban river corridors provide residents with spaces for recreational activities,

expanding opportunities for residents to connect with nature, thereby promoting their physical and mental health [17–21]. Urban river corridors have evolved into ideal areas featuring water landscapes, serving as venues for residents' social activities and community gatherings, fostering social cohesion among neighbors [22–24].

Over the past few decades, urban river corridor projects have been ubiquitous in developed countries and increasingly promoted in developing countries. A nuanced comprehension of public usage patterns and limitations is pivotal for effective urban river corridor restoration [25]. Residents inherently constitute a crucial component of urban river initiatives [26,27]. Identifying stakeholders' landscape values and preferences mitigates potential conflicts and promotes comprehensive planning and river corridor management [28]. Despite a conducive social and political milieu in China supporting heightened river landscape management and restoration for community well-being, public participation remains circumscribed, constraining substantive involvement from key stakeholders such as proprietors, users, residents, visitors, and neighbors. This limited public involvement dramatically hinders the effective bridge of the "people–city–river" relationship [28,29]. Empirical evidence underscores the detrimental consequences of neglecting public involvement in river restoration, particularly within urban river corridors [26]. Furthermore, involving the public in the decision-making processes of river restoration plans enhances their sense of belonging, ownership, and pride in the local river environment [30,31]. Ultimately, public participation in river projects heightens the probability of successful implementation and garnering support for restoration plans [25,32].

In the era of big data, exemplified by social media platforms, online networks have evolved into a domain where users can express opinions and emotions [33]. Platforms such as Facebook, Twitter, Flickr, Sina Weibo, WeChat, and Dianping boast substantial user bases, enabling individuals to articulate their viewpoints freely and discuss specific topics. Consequently, these platforms are evolving into crucial sources for collecting social perception data [34]. In contrast to traditional survey methods, the freedom, openness, and sharing reflected in web text data can authentically portray users' perceptions and experiences regarding urban green spaces [35–39]. The diversity and comprehensiveness of data on these social networks capture dynamic features of various human activities and environmental elements, providing researchers with significant convenience and rich possibilities to conduct urban studies from various scales and perspectives [40].

Some studies have attempted to explore the latent worth of user-generated reviews and tweets to enhance comprehension of recreational motivations and experiences [41,42]. Researchers often employ high-frequency word analysis and semantic network analysis to gauge visitors' preferences. For instance, Shen and Zhang [43] employed web text analysis using platforms such as Baidu, Dianping, and Ctrip to take comments and travel notes as samples. From the perspectives of environment, activities, attractions, services, and culture, they analyzed tourists' perceptions of wetland park tourist images. Some scholars have been experimenting with methods to transform information from spontaneously generated text data on social media into structured landscape feature assessments [44,45]. For example, Sim and Miller [46] collected 3703 tweets mentioning Jingyi Line Forest Park from Twitter to gain insights into the activities park visitors desired and their satisfaction levels with different activities. Plunz et al. [47] harnessed geographically tagged Twitter databases to develop a method for quantifying emotional levels in urban green spaces. Using Twitter and Weibo data as samples, Roberts [48] investigated seasonal variations in crowd activities and public engagement in urban green spaces. Based on Sina Weibo data, Fang et al. [49] also analyzed visitor sentiment characteristics in urban parks. They employed sentiment value calculation, word frequency analysis, and GIS spatial analysis and examined visitor spatiotemporal distribution characteristics, emotional structures, and key influencing factors. Upon organizing and analyzing these human-oriented activity data, the derived outcomes can be applied to the planning process, alleviating the deficiency in public participation within the landscape design process to a certain extent. While extensive research has explored online social media data in green spaces, its full potential in urban

river corridor research remains untapped. Information about the recreational experiences of urban river corridors awaits further exploration and discovery.

Understanding the interplay between individuals and urban river corridors is crucial for enhancing tourists' recreational experiences. This study integrates social-ecological models with qualitative analysis based on social media data. The theoretical foundation of social ecology focuses on individuals, emphasizing the intricate multilevel interactions between people and their environments [50]. It underscores the interrelationships between social and physical environments and individual behaviors; the actions of individuals, groups, and organizations can also influence social and natural environments [51]. As Sallis et al. [52] proposed, the social-ecological model provides a comprehensive theoretical framework for comprehending the multifaceted factors influencing the visitor experience in urban river corridors. A profound understanding of visitors' patterns of use and constraining factors in urban river corridors assists managers in comprehensively understanding visitor needs and expectations, thereby facilitating the provision of higher-quality urban river corridor services that align with citizen expectations.

Based on a qualitative ecological perspective, this study aims to explore the social and physical factors that promote or hinder recreational experiences along urban river corridors. By integrating the advantages of extensive text data mining, the research comprehensively and rapidly understands the environmental conditions and visitors' recreational experience needs, providing valuable insights for future urban landscape planning and management. The study focuses on three characteristics: (1) a preliminary evaluation of the recreational experiences along urban river corridors using high-frequency word features to induce the centrality of the semantic network within the text; (2) construction of a text-based topic model using LDA–Gibbs to determine the optimal number of themes and describe each theme; (3) content analysis of collected text data to identify visitors' recreational experience patterns and barriers they faced along urban river corridors.

## 2. Methods

### 2.1. Study Site

This study was conducted in the urban section of Hutuo River, Shijiazhuang, a city in northern China (Figure 1). Historically, this river has played a pivotal role in trade and agriculture, serving as the lifeblood for Shijiazhuang residents and encapsulating the city's rich history and cherished memories. In the last century, issues such as deteriorating water quality, sedimentation, interrupted flow, loss of biodiversity, and a decline in the river's landscape quality arose due to urban development and groundwater extraction in this river. The 2007 Hutuo River Flood Control and Comprehensive Improvement Project rejuvenated its landscape, combining aesthetic and natural landscapes, complemented by various recreational amenities, including jogging paths, dedicated bicycle lanes, dining areas, plazas, and a fountain hosting evening performance. According to the Urban Master Plan of Shijiazhuang City (2011–2020), the objective is to transform the Hutuo River Corridor into a green ecological corridor, aligning with the city's transition towards a "one river, two banks" layout [53].

### 2.2. Research Procedure

This study aims to conduct an in-depth analysis following the text mining of social media data related to the Hutuo River Corridor in Shijiazhuang, China. The research progresses through various stages, including data cleaning, feature word research, text theme model construction, and content analysis, with a step-by-step approach to delve into the core of the data. The specific processes involved (Figure 2):

(1) Cleaning the raw dataset obtained from the Dianping website.
(2) The LDA–Gibbs text theme model was employed to calculate the distribution of high-frequency feature words in the dataset, facilitating its structured processing. Subsequently, the perplexity of the overall text was computed to determine the optimal number of themes. A second round of LDA–Gibbs model computation was

performed based on the determined optimal theme number, resulting in the final theme text model.

(3) The Nvivo software (v.12) was employed to complete the coding of the text theme model, and the text content was analyzed to deeply explore the usage patterns and the constraints they encountered during their recreational experiences along urban river corridors.

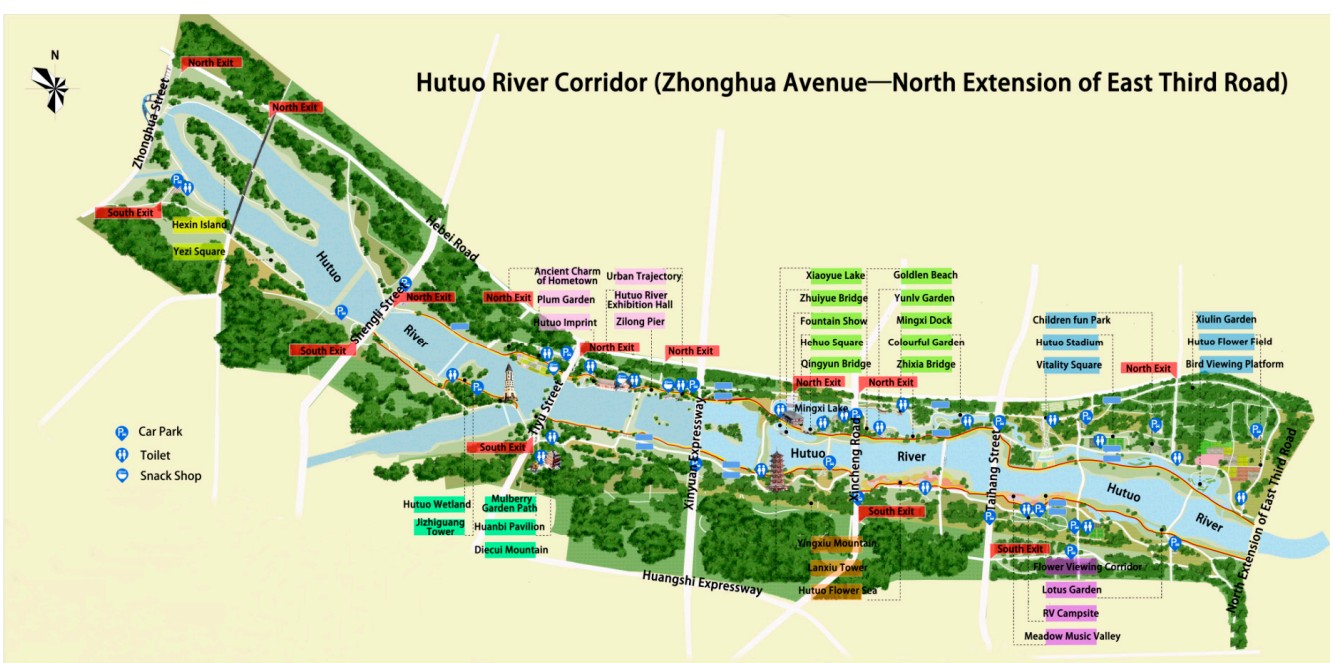

**Figure 1.** The urban section of Hutuo River Corridor in Shijiazhuang (Zhonghua Avenue–North Extension of East Third Road).

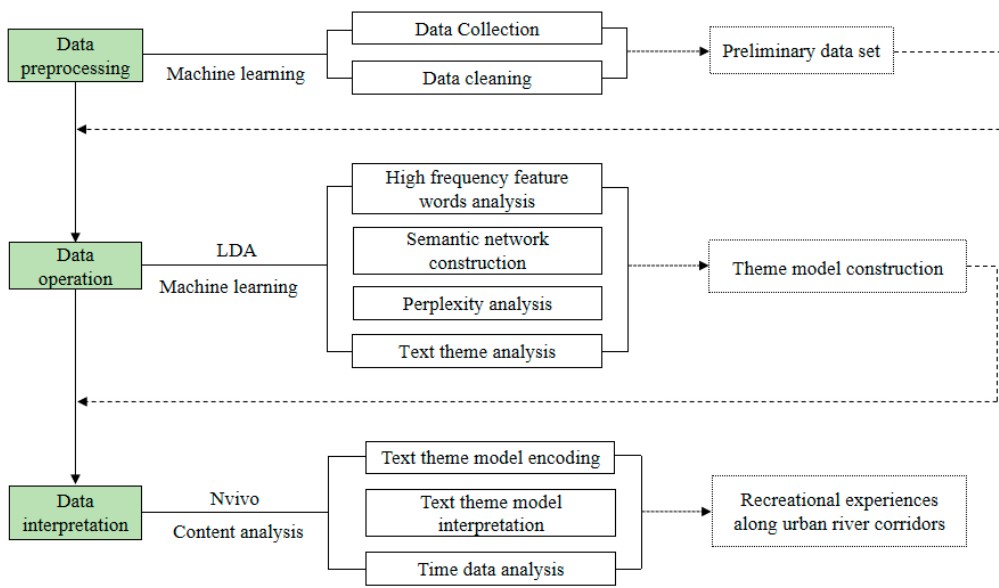

**Figure 2.** Social media data processing flowchart.

### 2.3. Data Collection and Cleaning

The research was conducted using ethical guidelines established by the JEthics Committee for Research Involving Human Subjects at Universiti Putra Malaysia (JKEUPM) [Ref. No.: 2023-1338]. Data utilized in this study were sourced from textual content (reviews)

available on Dianping (v.11.17.4), a prominent local life information provider and online rating platform in China. Dianping holds the second position in monthly active users according to the "2022 China's Top 10 Lifestyle and Leisure App Monthly Active Users Ranking" report [54], boasting a substantial user base spanning diverse socioeconomic backgrounds and age groups. All data collected from Dianping were voluntary submissions by users and publicly available, devoid of personally identifiable information. Consequently, users implicitly consented to using their data for research upon registering on the platform. The absence of encryption facilitated efficient data cleaning and analysis, supporting the extraction of valuable insights through algorithms. The utilization of Dianping data aligns with China's regulations governing data usage for academic research. Given the platform's representativeness, open-source nature, cost-free availability, and widespread usage across social media platforms, it emerged as the preferred source for foundational data in this study.

During the data collection process, measures were implemented to safeguard against the inclusion of sensitive or identifying information. Data were securely stored and accessible solely to authorized researchers. Users' comments underwent anonymization prior to analysis to ensure privacy and confidentiality. The final collected data structure encompasses the following variables: user nickname, comment content, review timestamp, and scenic spot name.

During data collection, it is possible to encounter dirty, noisy, or missing data. Therefore, performing preliminary cleaning while collecting the data, then the dataset was exported to Excel after collection completion, and multiple rounds of manual cleaning were performed based on the following principles:

(1) Exclusion of garbled data, such as emoticons or web links embedded within comments.
(2) Removal of meaningless characters like "!#¥%……Y#!!¥(*&".
(3) Elimination of duplicate data, where if a data entry appears more than once, only one instance is retained.
(4) Removal of invalid reviews, defined as those containing fewer than five characters or consisting solely of emoticon symbols such as "like", "good", or "okay".
(5) Elimination of data redundancy, excluding repetitive content within reviews, such as "good, good, good", "not bad, not bad, not bad", "like like like", and "recommend, recommend, recommend".

Given the revitalization and improvement of the Hutuo River (urban section of Shijiazhuang) that occurred after 2017, the data collection period spans from 1 January 2017, to 31 December 2023. After multiple meticulous cleaning, a total of 3006 valid review entries for the Hutuo River Corridor (urban section of Shijiazhuang) were obtained.

### 2.4. Data Analysis

2.4.1. LDA–Gibbs Text Theme Model Construction

This study conducted text mining on a preprocessed social media dataset based on the Latent Dirichlet Allocation (LDA). Natural language text was transformed into structured feature text. The LDA–Gibbs model was employed for semantic model construction and thematic analysis, including determining the number of themes and exploring their content.

The LDA–Gibbs model, introduced in 2003 by Blei and others, is a three-layer Bayesian probability model that is a powerful tool for analyzing the theme distribution in large textual datasets [55,56]. The fundamental concept is to represent documents as random mixtures of latent themes, each characterized by a distribution over words [55]. It provides a probabilistic representation of the themes in each document within a document collection. Analyzing a subset of documents and extracting their theme distributions makes it possible to perform theme clustering or text classification based on these distributions. LDA is an unsupervised machine learning method, eliminating the need for labeled data during model training. It is well-suited for selecting representative samples from vast datasets, classifying them, and annotating them [57]. Currently, LDA–Gibbs is widely applied in text mining, theme analysis, and related fields due to its stable and accurate results during the modeling process [58].

(1)    High Frequency Feature Word Extraction

In contrast to English words separated by spaces, the Chinese written language arranges characters individually without apparent gaps between words. To process this information, segmentation of the sequence of equally spaced Chinese characters is necessary for word separation [59]. Thus, a word segmentation tool is required to process the obtained text data, breaking down the original continuous text into word sequences and filtering out words with no practical meaning, i.e., stop words. This study used the Jieba segmentation tool to remove stop words from the Dianping text data. Subsequently, a word frequency analysis was performed on the text data to extract high-frequency feature words and generate a vocabulary frequency table. To avoid ambiguity, this study did not include the frequency of single characters in the statistics of high-frequency feature words. The top 100 high-frequency feature words were selected to gain initial insights into visitors' perceptions and the importance of urban river corridors. Subsequently, irrelevant entries were manually removed while concurrently retaining high-frequency terms that appeared together. The top 50 high-frequency feature words were selected to generate the semantic network relationship diagram. This diagram allowed for the analysis of node centrality, providing a more intuitive reflection of the semantic structure and interrelationships among high-frequency feature words.

(2)    Determining the Number of Themes

Empirical studies confirm a direct relationship between the effectiveness of LDA theme extraction and the number of latent themes, denoted as *K* [60]. The results of this extraction are susceptible to the value of *K*. Perplexity, a standard evaluation metric in natural language processing [61], which plays a crucial role in assessing the quality of models. Generally, perplexity decreases with an increasing number of themes, with smaller perplexity indicating more vital generalization ability. The "elbow" point, representing the optimal number of themes, exhibits a significant perplexity difference with the preceding point and a minor difference with the subsequent point [62].

This study employed the method of calculating perplexity and plotted the theme-perplexity curve to determine the optimal number of themes. In the LDA–Gibbs theme model, the formula for calculating perplexity is as follows [55]:

$$Perplesity(D) = \exp\left\{ -\frac{\sum_{d-1}^{M} \log p(w_d)}{\sum_{d-1}^{M} N_d} \right\} \tag{1}$$

where, *D* represents the test set in the corpus, consisting of *M* documents. $N_d$ denotes the number of words in each document *d*, $w_d$ represents the words in document *d*, and $p(w_d)$ represents the probability of generating the word $w_d$ in the document.

(3)    Construction of Text Theme Model

After determining the optimal number of themes, *K*, through perplexity calculation, a random assignment of a theme number was made for each word in the corpus. Within each theme, words were arranged based on their probabilities of generation. The distribution of words in each document and the associated themes was computed, resulting in the document–theme distribution. Throughout this process, careful attention was given to the issue of polysemy, where feature words needed to appear in each corresponding context when such instances arose. This procedure involved organizing and categorizing the word frequency results, ultimately establishing categories for textual themes.

### 2.4.2. Text Content Analysis

Next, Nvivo software was employed to conduct content analysis on the comment data within the social media dataset. The key method practically interprets themes within the LDA–Gibbs model by integrating comment texts and high-frequency feature words through theme analysis to provide insights into visitors' attitudes and thematic interests concerning the urban river corridor. The overall process is outlined as follows:

(1) Data import: select the top hundred comments most relevant to each theme and import them into the Nvivo folder.
(2) Contextualization of feature words: combining feature words generated by the LDA–Gibbs model and comment data from social media, the feature words are contextualized.
(3) Manual coding: interpreting the results obtained through Nvivo analysis involves summarizing the specific content described by each theme.
(4) Output of results: after completing the node classification, specific descriptions of each theme can be obtained.

### 2.4.3. Time Data Analysis

This section analyzes the vibrancy of urban river corridors based on a time series, considering seasons, months, weekdays and weekends, holidays, and specific times of the day when visitors engage in activities [63]. The COVID-19 pandemic pronounced impacted the commentary volume related to the Hutuo River Corridor in Shijiazhuang, given the home isolation measures imposed on residents during the outbreak. Therefore, the analysis centers on data gathered post-lockdown from 1 January 2022, to 31 December 2023. The study examines the monthly comment on the Hutuo River Corridor in Shijiazhuang and the comment volumes during four seasons: spring (March to May), summer (June to August), autumn (September to November), and winter (December to February). The aim is to depict the current state of visitor activity and experiences accurately.

## 3. Results

### 3.1. High-Frequency Feature Word Analysis

In accordance with the data format obtained, high-frequency keywords were extracted using the LDA–Gibbs model, resulting in an original Chinese word segmentation set. Subsequently, data from the original Chinese word segmentation set underwent filtering, utilizing a predefined stop-word list to obtain an adjusted textual dataset. This process involved removing irrelevant vocabulary and merging duplicate words, culminating in extracting 100 high-frequency terms related to the urban section of the Shijiazhuang Hutuo River Corridor (Table S1).

From a lexical perspective, high-frequency vocabulary primarily comprises nouns, verbs, and adjectives. Among these, nouns constitute the most significant quantity, encompassing visitor types, geographic locations, scenic spots, facility names, functions, and visitation times. Verbs chiefly reflect features of visitors' behaviors, motivations, and the process of visitor activities. Adjectives are mainly employed to portray visitors' attitudes, moods, and perceptions and describe the modified architectural image features, reflecting the sensory characteristics of the Hutuo River Corridor. The adjective "good" stands out as the most frequently mentioned term, indicating overall satisfaction with the urban river corridor. "Children" is the most mentioned noun, suggesting that the Hutuo River Corridor provides an ideal place for family outings for residents. Additionally, visitors may access with "friends". Nouns like "park", "Hutuo River", "Shijiazhuang", "scenic area", "Zhengding", "riverside", "flower sea", "nearby", "location", "Taiping River", and "scenic area" represent visitors' understanding of the geographic location of the scenic area. Other nouns such as "time, scenery, environment, lighting, weather, flower sea, facility, transportation, season" indicate visitor interest in the urban river corridor's environmental features and physical attributes, with a preference for visiting during favorable weather. "Tickets" and "free" rank high, underscoring visitor concern about the corridor's fees and related charges. Adjectives like "suitable, beautiful, convenient, attractive, happy, fun", along with "recommendation" and "worth" reflect visitors' experiences. Verbs such as "perceive, check-in, drive, like, come out, take photos, barbecue, camp, drive, play, picnic" express visitors' deep engagement in experiential behaviors during visits, highlighting the corridor's diverse offerings and functions. However, the "parking" issue may leave room for improvement.

A semantic network graph was constructed using the top 50 high-frequency feature words (Figure 3), followed by an analysis of node centrality using degree centrality. The size of nodes represents their degree of centrality, with larger nodes indicating higher centrality, positioning them as pivotal elements within the network. Edge thickness signifies the strength of association between two nodes, where thicker connections denote more co-occurrences and stronger correlations between words.

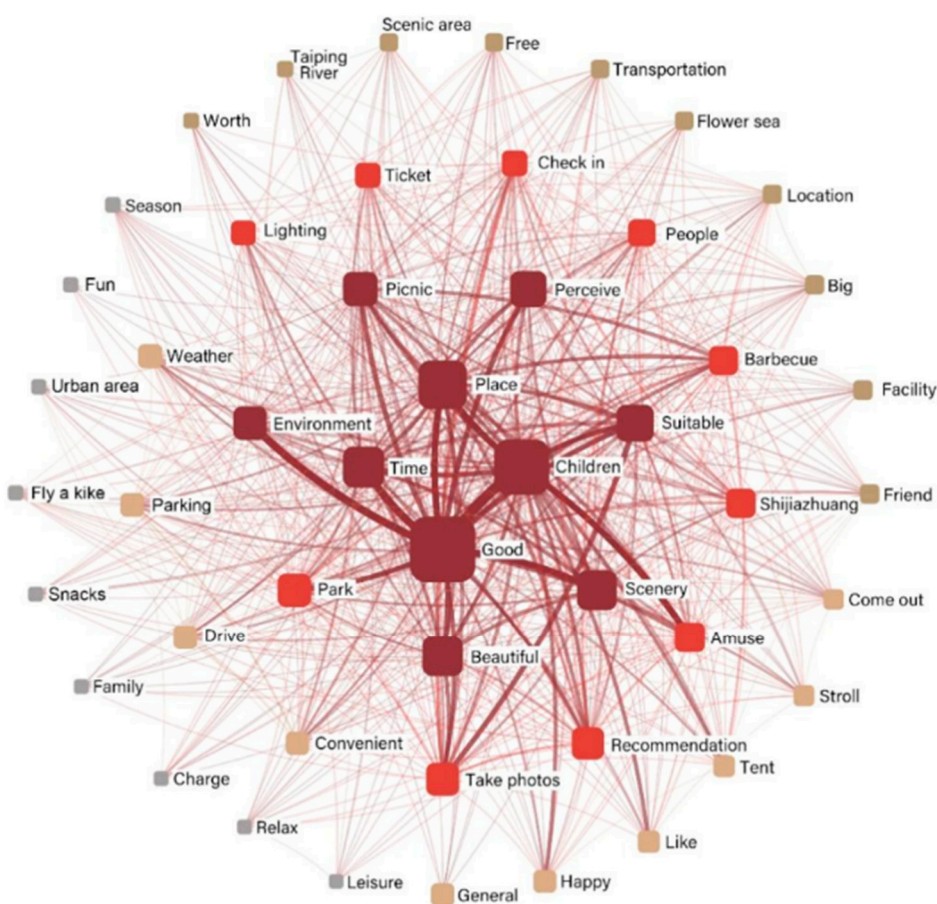

**Figure 3.** Semantic network centrality of social network texts in Hutuo River Corridor.

Figure 3 exhibits a divergent pattern radiating from the center outward. "Good", "children", and "place" emerge as the predominant high-frequency words, often serving as the focal points for evaluations. This phenomenon epitomizes a concentrated reflection of urban river corridor utilization. Among these, "children" holds a central position, serving as the most critical node in the network. It is frequently associated with vocabulary expressing children's activities, locations, and moods, such as "good, take photos, place, amuse, scenery, suitable, happy, beautiful", and more. Elements related to the destination, centered around the core term "place", form prominent themes in the comments, including "perceive, good, children, scenery, park, time, check-in, barbecue, beautiful". These elements exhibit a closely interconnected structure. Second-level keywords encompass time, activity locations, and experiential terms, providing further insights into the core positions. Visitors' activities like "take photos, barbecue, check-in", along with nouns such as "lighting, ticket, people, Shijiazhuang, park", and adjectives related to experiential feelings collectively constitute third-level keywords. Additionally, concerns about "weather, parking, drive" are evident. Expressions like "amuse" and "convenient" indicate visitor satisfaction with the urban river corridor environment. The outermost and peripheral positions represent further expansion and enrichment of the core and sub-core positions. Through this four-tier structure, the attractor system of the urban river corridor is relatively complete, providing a profound understanding of the interactive dynamics among visitors, destinations, and activities. This

framework opens new perspectives for enhancing the image and constructing a future system for urban river corridors.

### 3.2. Text Theme Model

Figure 4 illustrates the perplexity function's output results for different themes. It is evident that perplexity gradually decreases as the value of *K* (number of themes) increases. According to the elbow method, a significant inflection point occurs around *K* = 8, where the decreasing trend of perplexity slows down. Therefore, the optimal value for K, in this corpus selected for our study, is identified as 8. Utilizing the LDA model, the generation quantity of LDA themes was set to 8.

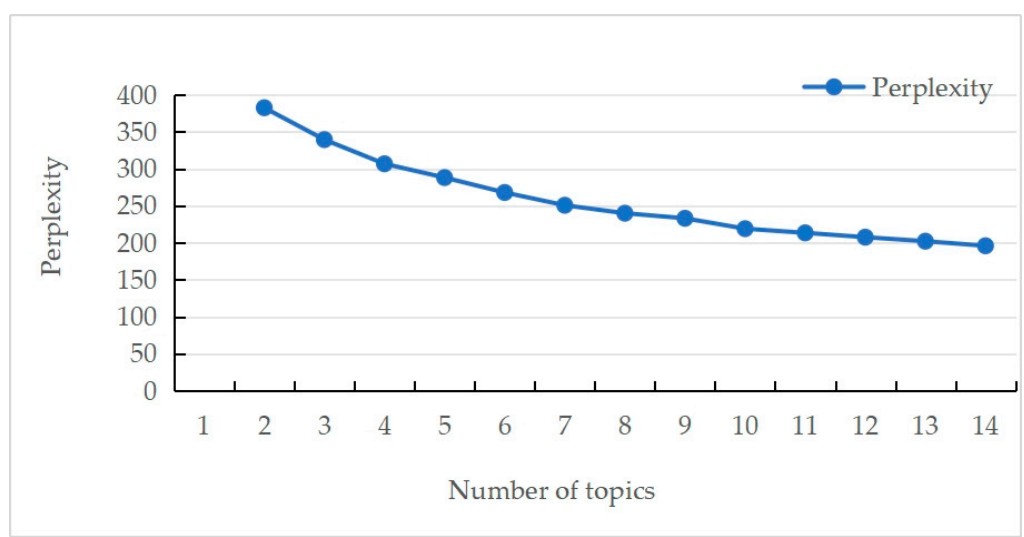

**Figure 4.** Number of theme–perplexity line graph.

Therefore, the results extracted from the LDA–Gibbs model encompass eight themes. For each theme, the feature words are sorted based on their generation probability within that theme. The top 30 words for each theme are selected and presented in Table S2. The LDA–Gibbs model randomly selects text themes and compares the similarity of words in the comments. After marking similar words, the model assesses the frequency of word occurrences. Simultaneously occurring high-frequency words are grouped into the same theme. In a given comment dataset, characteristic words from different themes may appear, with the proportion of high-frequency words determining the thematic attributes of that comment. By drawing inspiration from the coding results of Nvivo and integrating comments with high-frequency feature words, the following themes were systematically summarized: visit time, visitor types, visit motivation, visitor activities, characteristics of urban river corridor, maintenance and management, accessibility, and visitors' attitude, as outlined in Table 1. Subsequently, a thematic overview was provided for the eight themes), accompanied by illustrative comment examples (Table S3).

**Table 1.** Summarized themes and categories.

| No. | Theme | Category |
|---|---|---|
| 1 | Visit time | Season, month, weekday and weekend, holiday, time of a day |
| 2 | Visitor types | Children, adults, the elderly, parents, friends, couples, etc. |
| 3 | Visit motivation | Admiring the scenery, exercising, spending time with family or friends, discovering and experiencing nature, relaxing, escaping city life, engaging in outdoor recreation and services |

**Table 1.** *Cont.*

| No. | Theme | Category |
|-----|-------|----------|
| 4 | Visitors' activities | Scenic appreciation, sports activities, interact with small animals, skiing, barbecuing, picnicking, drinking, conversations, dancing, dog walking, walking, etc. |
| 5 | Characteristics of urban river corridor | Natural landscape (vegetation, animals, water), artificial facilities (fitness trails, sports facilities, fountain, outdoor terraces, pavilion, sculpture, etc.) |
| 6 | Maintenance and management | Maintenance, management, services |
| 7 | Accessibility | spatial openness, continuity, amenities, landmarks, traffic conditions, distance, and models of transportation |
| 8 | Visitors' attitude | positive, neutral, and negative |

### 3.3. Factors Affecting Leisure Experience in Urban River Corridors

The present study integrates the eight themes identified by the LDA–Gibbs model with high-frequency words, Nvivo coding, and textual comments. Through extensive text reading, contextual relationships among them were unveiled and reconstructed. Simultaneously, based on the social-ecological model, factors influencing the recreational experiences along the Hutuo River Corridor in Shijiazhuang were reorganized into two major categories: social and physical attributes (Figure 5). Social attributes refer to the individual's social psychological processes connecting with others, society, and nature in social practices [64,65]. The research indicates that leisure experiences along the urban river corridor are primarily influenced by social factors such as visit time (weekdays and weekends, holidays, and time of day), duration of stay, motivation, safety, and visitors' types and activities. Meanwhile, physical attributes encompass the tangible features and spatial elements of objects [66]. Physical factors influencing the recreational experience along the urban river corridor mainly involve natural elements, artificial facilities, maintenance and management, accessibility, distance, and models of transportation. Although weather and seasons (months) are related to time, they often cause variations in the landscape environment, affecting visitor access, and are thus considered physical factors.

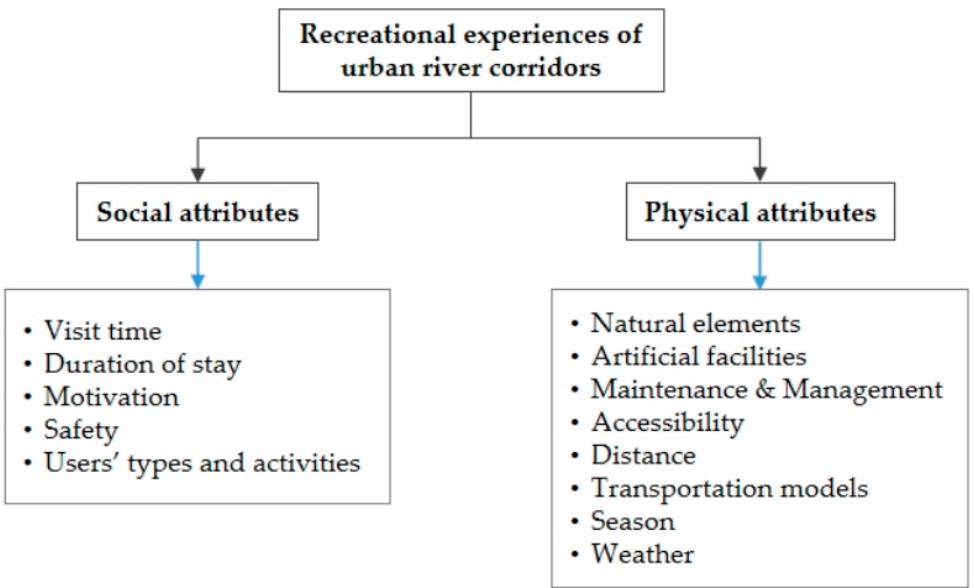

**Figure 5.** Factors influencing recreational experiences of urban river corridors from socio-ecological perspective.

### 3.3.1. Social Attributes Influencing Recreational Experiences of Urban River Corridors

(1)　Visit time

Temporal variations significantly influence the visitation to the Hutuo River Corridor in Shijiazhuang. In terms of social attributes, it is mainly analyzed and discussed from three aspects: weekdays and weekends, holidays, and time of day.

The comments indicate that a surge in user numbers is observed during weekends and holidays. Many visitors mentioned that the Hutuo River Corridor is an "excellent destination for weekend outings" and noted that "almost every weekend, the place is crowded". Residents prefer leisure time in nature [67]. Weekends and holidays offer most people a break and leisure, giving them more free time to visit and explore scenic spots. Weekdays may be constrained by daily tasks such as work and study, particularly for children who, influenced by China's unique education policies and competitive mechanisms, face heavy homework loads. As supervisors and companions for children, parents also find it challenging to allocate time for recreational experiences in urban river corridors during weekdays [67,68]. Additionally, many special events and entertainment activities typically occur on weekends and holidays, attracting more participants.

Diverse visitation times throughout the day showcase the corridor's capacity to attract foot traffic [69]. According to high-frequency feature words, "evening" is frequently mentioned. Visitors access during the evening in various seasons; in summer evenings, there is a high influx of people seeking relief from the heat and enjoying the night scenery. Common sentiments include, "There are usually more people there", and "It is cooler by the river, a great place to bring the whole family for a refreshing break". Evening light shows receive praise, with visitors marveling at the "colorful lights creating a magnificent scene as if entering a fairyland". In winter evenings, organized events attract people, such as, "On the fifteenth night of the first lunar month, drove to the spacious Mingxi Lake Park to enjoy fireworks and sky lanterns". However, a visitor advisory notes, "It can be a bit cold in the evening, so remember to dress warmly". "Midday" appears with moderate frequency. In May, visitors choose to "nap lightly in the tent at noon" or "take a nap on the grass". Towards the end of October, "the noon sunlight falls on the sand, and children play barefoot". However, during summer and early autumn, as it can be "too hot", it is "not recommended to visit during the heat of midday". Although "morning" has a lower frequency, it is recommended by many. Residents in the vicinity often come for morning jogs, and sightings of fishing enthusiasts are common. Expressions like "A morning stroll here is very comfortable and pleasant" reflect that the morning offers a serene and inviting environment, attracting numerous fitness enthusiasts.

(2)　Duration of stay

The majority of findings suggested that visitors' duration of stay along urban river corridors is typically short, approximately one hour [70–76]. However, this study reveals diverse visitor stay duration. Some visitors merely pass through, taking a moment to "take a deep breath, let thoughts briefly linger, and savor the tranquility of life". Others describe, "Choosing a day with moderate temperatures, finding a patch of grass, laying down a mat, bringing some food and drinks, one can camp here for an entire day".

(3)　Motivation

Human preferences play a crucial role in supporting positive experiences and fulfilling their motivation for visiting a site [77]. Based on the textual content, visitors' motivations can be categorized into six distinct types: admiring the scenery, exercising, spending time with family or friends, discovering and experiencing nature, relaxing, and escaping city life, as well as engaging in outdoor recreation and services.

Most visitors come here to admire the scenery, as reflected in the comments: "Always seeing other people's pictures in the comments, today I finally got to see the beautiful scenery, truly beautiful!" and "The setting sun on the water, a gentle breeze brushing the face, blue sky with white clouds, and swaying reed beds". This inclination towards

appreciating natural beauty indicates people's pursuit of aesthetically pleasing moments. Furthermore, visitors express satisfaction and joy when enjoying the picturesque views of the Hutuo River Corridor: "The surrounding views, such as flowing water and lush greenery", seem to evoke people's experiences of tranquility and the beauty of nature. Overall, the river corridor attracts leisure and entertainment seekers and serves as a refuge for those who appreciate and draw inspiration from natural landscapes. This shared appreciation creates positive and enriching experiences for individuals frequently visiting this natural scenery.

Furthermore, visitors access this place to discover and experience nature. Research in environmental psychology indicates that vegetation and water bodies in natural environments contribute to stress reduction, enhanced attention, and positive emotional impacts [78]. In natural landscapes, vegetation provides visual pleasure and creates a pleasant environment for visitors, fostering mental health and emotional well-being [79]. On the other hand, the significance of water in shaping people's perception of nature is underscored. When combined with natural water features such as grassy embankments, they have been found to contribute to a sense of tranquility and encourage enjoyable activities like contemplation or sitting down to observe, aligning with people's preferences for nature experiences [80,81]. The broad water body of Hutuo River, earning widespread acclaim from citizens, provides an enhanced panoramic view. Various nature experiences mentioned in the comments, such as "sitting by the river", "feeling the breeze", "admiring flowers and plants", "listening to bird calls", "hearing the sound of water", and "breathing fresh air", reflect the avid pursuit of natural elements by individuals. Particularly under the influence of the pandemic, the heightened demand for outdoor spaces highlights a desire for sunlight and fresh air. The Hutuo River corridor, as an urban green space offering abundant vegetation and water features, satisfies citizens' aspirations for nature, tranquility, and a healthy lifestyle.

Moreover, the design of the Hutuo River Corridor draws individuals to engage in physical exercise. Adequate physical activity has been shown to mitigate the risk of diseases associated with urban living, such as diabetes, hypertension, and cardiovascular diseases [82–84]. The Hutuo River Corridor is meticulously fashioned as an enclave for physical activity, featuring dedicated fitness trails, exercise equipment, and recreational areas. For instance, one user comment eloquently states, "A new running track has been established here, serving as a natural oxygen bar for sports and fitness". The Hutuo River Corridor emerges as a community resource propelling a healthy lifestyle, offering residents a space conducive to promoting both physical and mental well-being.

Chinese residents exhibit a profound sense of relationship [67], with many considering the enjoyment of social relationships, such as spending time with family or friends, as a significant aspect of their lives. The Hutuo River Corridor, as a public space, evidently serves as a popular venue for shared moments with family or friends. Various activities like "family gatherings", "car enthusiast meets", and "company team-building" choose this location. "Having a picnic here when the weather is good is also delightful". "On weekends, inviting a few good friends to camp here, savoring red wine, enjoying barbecues, singing songs—it is all about comfort and happiness". Numerous visitors echo these sentiments. Perceiving the environment, engaging in conversations, and utilizing the river corridor as a recreational space or sitting by the riverbank to savor culinary delights all contribute to shared activities, imbuing the setting with symbolic meaning. The resulting social networks, friendships, and relationships contribute to collective well-being [85,86].

In contemporary urban settings, alleviating daily stress has become increasingly crucial. One study indicates that, unlike artificial environments, natural settings can restore physical vitality and offer mental relaxation [87]. Consequently, some users regard the Hutuo River Corridor as a place to "escape the hustle and bustle of the city, unwind, and rejuvenate". Another visitor comment elucidates, "In the urban area, the Minxin River that runs through the city fails to fulfill people's desire for a waterside residence in terms of water quality and scale. The ecological project of the Hutuo River may bring a sense of

anticipation to those long accustomed to city life. While it lacks the exquisite and enduring beauty of southern water towns or the grandeur of mighty rivers, it still imparts people a serene joy and tranquility".

The Hutuo River management authorities also offer various outdoor leisure activities and services. Unlike urban areas in the UK, in Shijiazhuang, it is challenging to find dedicated dog parks. Consequently, many residents opt to walk their dogs along the Hutuo River. For those with a penchant for fishing, "the river environment is quite clean, teeming with fish, making it an ideal spot for fishing". Providing activities within this space enhances opportunities for public participation, thereby promoting the utilization of open spaces [88]. For instance, some individuals visit the Hutuo River to engage in beer festivals, light displays, concerts, fountain performances, boating, and winter activities like skiing with their children. The site encapsulates beliefs, identities, memories, and thoughts that transcend everyday concerns. For instance, some come to the Hutuo River Corridor to rediscover childhood tastes, noting the presence of snack vendors, sugar painting artists, cotton candy, and old-fashioned popcorn, which are increasingly rare in the city. The creatively designed Green Train Head restaurant, "laden with locals from Shijiazhuang, evokes childhood memories".

(4)   Safety

In this study, user comments primarily highlight two safety concerns. Firstly, the narrow stairs inside the Lanxiu Tower may increase the risk of accidents. Secondly, the issue of haphazard parking could lead to traffic chaos and jeopardize pedestrian safety. These problems may influence visitors' perceptions and experiences of the Hutuo River Corridor, underscoring the need for careful consideration and resolution to enhance the overall quality of public space.

(5)   Visitors' types and activities

In contrast to prior research, this study reveals a surprising finding—the most frequently mentioned visitors in terms of frequency are children. Many comments center around children's activities: "The favorite activity with my children is rowing in the park", "Many kids are playing in the water by the river", "Children by the river are busy digging sand and catching fish, having a great time". Children's playtime is primarily determined by the structure of the school year, with summer vacation being their preferred time for outings [89]. Warmer seasons are more conducive to physical activities for preschool children, while colder seasons pose more significant challenges [89]. Consistent with a study in Finland, outdoor activities for children often require supervision [74]. Therefore, some considerate visitors advise parents with children to be cautious and keep a safe distance from the water. Additionally, young adults constitute a significant visitor group, including parents of children. Some engage in activities organized along the Hutuo River Corridor with friends, attending concerts, light shows, and other performances, attracting mainly young people and couples. The enchanting scenery of the corridor also draws many young couples for wedding photoshoots. Elderly visitors often come for a stroll, accompanied by family members or walking their dogs, which is consistent with previous studies [90,91]. Some seniors specifically come for fishing, while others participate in morning exercises, contributing to the diverse visitor demographic of the Hutuo River Corridor.

Scenic appreciation is the most frequently mentioned activity by user comments, including enjoying flowers, autumn landscapes, river views, and watching sunsets. For instance, during spring, "the flora along the river flourishes", particularly along the "entire flower-lined route, creating a liberating and delightful ambiance". "The river water is clear, rippling with a picturesque springtime scenery". In summer, "the lovely wetland park built along the river features vast lotus ponds". As autumn arrives, "a beautiful scene unfolds with blue skies, white clouds, green water, and yellowing grass". Adults prefer springtime outings and photography due to the rich vegetation and vibrant colors. One comment noted, "In places with dense trees, many people are sketching".

Halkos et al. [92] suggest that infrastructure can enhance visitor satisfaction, especially with sports and recreational facilities [93]. Many people engage in sports activities here because "the fitness trail along the river has a comfortable feel", and "many people walk and run here on weekends. There are also quite a few cycling enthusiasts". "Climbing on Wohushan Park, a small pavilion is at the top. Although not high, it provides a panoramic view of the mountains!" The Hutuo River Corridor also has numerous well-designed and safe facilities for children, offering water activities, sand play, boating, kite flying, skateboarding, and balance biking. It is a professional and secure space, highly suitable for family entertainment. Some parks with animal themes allow children to interact with small animals. In winter, many parents deliberately bring their children here to ski. When the riverbank freezes, some even consider it a natural ice-skating rink.

Research indicates that positive acoustic environments in urban green spaces are perceived in various ways, contributing to tranquility and peace during outdoor leisure experiences [94–97]. For instance, the sound quality of flowing water can be utilized by urban planners to create a soothing soundscape [98]. Furthermore, the deliberate emphasis on ecological conservation features in the Hutuo River Corridor attracts birds to nest. A user commented, "Lying in a hammock, feeling the gentle breeze, and listening to the birdsong is truly delightful".

Moreover, young people refer to the Hutuo River Corridor as a "great camping spot". They often set up tents with family or friends, bringing small tables, chairs, grills, and meat. They engage in activities such as barbecuing, picnicking, drinking beer, and engaging in lively conversations, referring to this place as a "gathering place for toasting". Additionally, some users observe that "many people gather by the river, some with children playing", similar to the concept of "child walking" in Finland [74]. "In the mornings, you can often see some elderly individuals, some practicing Tai Chi, others gracefully dancing with fans". "There are also those who come here to walk, and others walking their dogs, and so on".

### 3.3.2. Physical Attributes Influencing Recreational Experiences of Urban River Corridors

(1)    Natural elements

Urban river corridors, as distinctive natural spaces within cities, play a crucial role in urban planning and ecosystem services [99]. Firstly, these corridors often exhibit rich biodiversity, functioning as green ecological corridors in urban settings [100]. For instance, "extensive planting of flowers and trees along the river has created a floral ocean"; in spring, "yellow rapeseed flowers, purple February orchids, white pear blossoms, and pink cherry blossoms create an exceptionally enchanting scene". In summer, "lush greenery prevails, and the most extravagant display is the vast floral sea, with a large area and diverse varieties, recreating the scenery of blue skies, green meadows, and reeds-filled ponds". However, due to the relatively short time since the Hutuo River restoration, some sections still lack fully grown trees, leading to complaints about "small trees, insufficient shade, and too much sun exposure". In autumn, admiration is expressed: "The abundant flowers, plants, and trees on both sides of the river create a dense and verdant landscape, with reeds swaying gracefully in the wind. Occasionally, water birds fly by, and along the roadside, the flower sea, though not as abundant as in midsummer, presents a unique scenery". Che et al. [101] proposed that vegetation contributes to people's experience of rivers and water environments. The diversity of the ecosystem also attracts various wildlife, contributing to essential ecological balance. Users remark, "More bird species are discovered here", "seagulls are sometimes seen on the water", "walking down the observation path towards the deeper areas leads into the reed marsh, where ducks can be spotted swimming or foraging occasionally", "there is quite a lot of fish", and "butterflies dance gracefully among the flowers". Secondly, these corridors typically revolve around water bodies, forming a natural water system network. This not only aids urban water resource management but also provides a comfortable leisure space. Nature-oriented river corridors attract people by meeting their need for contact with natural water [102]. Börger et al. [103] found that water quality may influence the leisure experience in urban river

corridor spaces. Additionally, their research indicates that water quality assessments may be based on visual cues, such as color, clarity, and the presence of debris [103]. Based on textual evaluations, users commonly express sentiments like "large water surface", "good water quality", "relatively clear water trail", "clean river water", and "the river is vast with a broad view", indicating high visitor satisfaction with the river water.

(2)    Artificial facilities

The Hutuo River Corridor not only possesses rich natural landscapes but also integrates various artificial facilities to meet the diverse needs of residents better, enhancing the overall functionality of urban green spaces. Firstly, it features dedicated fitness trails and sports facilities, catering to residents' needs for physical exercise. These facilities offer convenient fitness spaces and create shared community areas, fostering a healthy lifestyle, such as cycling for the youth, post-dinner strolls, and Tai Chi for the elderly. Secondly, artificial landscape designs like the "musical fountain", the "I Love Shijiazhuang" sign, colorful windmills, and wind chimes embellish the route, adding artistic elements and aesthetic appeal. These elements enhance the city's visual appeal and provide residents with places to appreciate and relax. Additionally, urban river corridors often include children's play areas, offering kids a safe and enjoyable space. Facilities like outdoor terraces become ideal places for family and friend gatherings, supporting community activities. People gather in these spaces, participating in various outdoor activities such as picnics and barbecues, making them an integral part of urban life. Places where people gather often attract commercial and entertainment establishments, enriching social activities and enhancing the site's allure [104]. Consequently, the Hutuo River Corridor features a dedicated area for water performances, injecting vitality into the riverside space and attracting more visitors and residents. Lastly, to enhance the visitor experience, some river corridors have rest pavilions, train-themed restaurants, and other service facilities, allowing people to enjoy the natural beauty along the riverbank better. Moreover, these convenient facilities carry rich cultural histories, reflecting the evolution of the city and traces of human activities. Intertwining these artificial facilities with natural features collectively constructs a unique and comprehensive urban ecosystem for river corridors.

(3)    Maintenance and Management

The maintenance and management evaluation primarily focuses on the green space maintenance and overall landscape management of the Hutuo River Corridor. Overall, visitors have provided positive feedback on the green space maintenance of the Hutuo River Corridor, noting that the upkeep is timely and issues such as overgrown weeds and garbage disposal have been effectively addressed. One user commented, "In my previous memory, both sides of the Hutuo River were filled with overgrown vegetation, litter everywhere, and dirty river water. Unexpectedly, in the past two years, the riverbanks have been transformed into a park with lush greenery, outstanding cultural and natural scenery, and the water quality feels excellent, with occasional sightings of people fishing". However, some users expressed dissatisfaction concerning the convenience facilities, stating, "The equipment is somewhat outdated, and we hope for enhanced maintenance". "In the Lanxiu Tower area, the hygiene maintenance is insufficient, lacking garbage bins. Tourists litter everywhere with garbage and cigarette butts. It would be even better if tourists could conscientiously contribute to maintaining the environment". Influenced by peak tourist periods, there is an increased demand for convenience facilities, leading to complaints such as, "The toilets are overcrowded, not cleaned promptly, and hygiene is poor". They suggested, "While the scenic environment is good, and the view is beautiful, the supporting facilities need to catch up". Regarding management, some areas require strengthening; as users pointed out, "For some fee-based services, the charges are deemed unreasonable". "Small vendors are selling somewhat unsanitary snacks on both sides of the roads". These factors are pivotal in shaping visitors' overall experience and satisfaction with the Hutuo River Corridor.

(4)　Accessibility

The accessibility of urban river corridors involves various dimensions. Previous research by other scholars has incorporated variables such as spatial openness, visual corridors, continuity, and amenities to assess the public's accessibility to urban riverbank spaces [101,105]. By extracting high-frequency keywords from social media data and summarizing textual content, this study identified five indicators influencing the accessibility of the Hutuo River Corridor: spatial openness, continuity, amenities, traffic conditions, and landmarks.

In certain circumstances, river corridors might be repurposed, impacting the convenience for the public. For example, comments indicate that the Hutuo River Corridor was intercepted during flood periods, and many areas were sealed off. "Due to water discharge in the river, the entrance gates have been locked". Such situations may restrict public access to the river corridor during specific periods or under particular conditions. Within user comments, three instances highlight disruptions caused by road construction, leading to detours. Such interruptions in continuity may significantly interfere with visitors' travel and activity plans, diminishing their willingness to engage with the space. In addition, visitor feedback indicates challenges such as "difficulties in traversing uneven road surfaces" and reports of "locked restrooms, rendering them inaccessible". These issues may pose inconveniences for the public when utilizing river corridors.

Users' comprehensive assessments of the transportation conditions along the Hutuo River Corridor indicate favorable perceptions: "The roads are relatively wide, with good traffic conditions and convenient transportation", and "Parking is free". Many users use self-driving to reach the Hutuo River corridor, as its expansive area allows flexible parking opportunities. As one user expressed, "You can park along the way, stroll wherever you would like, and stop whenever you desire". Typically, "parking spaces are abundant" during regular times. However, during peak seasons, parking resources may fall short of meeting the heightened demand, leading some users to remark on the challenges of overcrowding: "Too many people, too many cars, making parking difficult". Therefore, comprehensive considerations of traffic flow and parking requirements are essential in planning and managing urban river corridors, ensuring visitors can enjoy convenient transportation experiences at different times.

Extraction of high-frequency keywords revealed numerous nouns related to signs and roads, including "Bridge", "Street", "Square", "Avenue", "Exhibition center", and "Landmark". Many user comments enthusiastically shared specific locations along the Hutuo River Corridor, offering valuable guidance to others. For instance, users provided detailed directions such as "Turn right after crossing the Zilong Bridge, continue along the scenic avenue, passing by many popular landmarks like the small train, beach, and flower sea. Further ahead, you will find the beloved landmarks of Shijiazhuang". Some users even clarified the significance of the "Jizhiguang Tower" by stating, "Situated in the northern outskirts of Shijiazhuang, the iconic 'Jiziguang' Tower stands prominently on the eastern high slope. The tower's design creatively incorporates the abbreviation of Hebei Province, the character 'Ji,' showcasing a marvelous conceptualization and unique style, reflecting rich regional cultural elements". These detailed location insights and route suggestions not only assist tourists in exploring the Hutuo River Corridor more effectively but also contribute to the overall accessibility of the region.

(5)　Distance

Giles-Corti et al.'s study revealed that distance influences whether and how residents access natural spaces beyond a certain threshold [106]. Numerous empirical studies from various countries have consistently demonstrated a decline in the frequency of visits to green spaces with increasing distance [107–109]. Within this study, user comments on the distance to the Hutuo River Corridor predominantly convey a sense of proximity, with expressions such as "the journey is not far", "approximately 10 km from the city center", and "a drive takes about 20 min". These remarks underscore the relatively close geographical location of the Hutuo River Corridor, enhancing residents' willingness and frequency of

visitation. This underscores the importance, in urban planning, of integrating natural spaces into the urban layout and ensuring their relative proximity for the convenience of a broader population to access and enjoy these natural environments.

(6)    Transportation models

In land use and transportation studies, transportation modes are defined as the individual's choice of the degree to which they adopt specific means of reaching activities or destinations [110]. Within user comments, there is a consensus recommending private car usage for greater convenience, with expressions like "driving oneself is more convenient" and "walking is also possible as the locations for each attraction are not too far apart". However, due to the relatively expansive area, walking might be perceived as tiring. Visitors also note potential traffic congestion during peak hours and the challenge of finding parking spaces. Although options like taking Metro Line 1 to the Convention Center or utilizing direct buses (77, 130, and 177) from the city center are available, they involve additional walking, and public transit options have early cut-off times. Consequently, public transit is deemed feasible but not ideal, especially during evening events when taxis are scarce. Some users suggest cycling if the distance is not too far, as it allows flexible parking and is not affected by traffic congestion. Cyclists are relatively common, and some prefer morning rides along the Hutuo River Corridor, emphasizing its suitability for a round trip of 20 km without disrupting daily routines. These diverse transportation comments underscore the importance of convenience, comfort, and flexibility considerations.

(7)    Season

After February, the steady growth in the volume of comments may reflect a seasonal trend in the Hutuo River Corridor. Figure 6 illustrates increased comments during the spring season, indicating a preference for visiting the corridor. The rise in temperature during spring and the allure of seasonal landscapes contribute to this trend. Prior research supports the notion that compared to colder months, individuals are inclined to engage in outdoor activities during warmer months [89,111]. Therefore, after winter, "with the warming weather, people are eager to step out of their homes, immerse themselves in nature, and enjoy the beauty of spring". As described in comment 1158: "With the arrival of spring, everything comes back to life. Green leaves sprout from the branches, and small flowers are about to bloom. The reeds by the river sway in the wind. Taking advantage of the good weather, come out for a stroll with the kids".

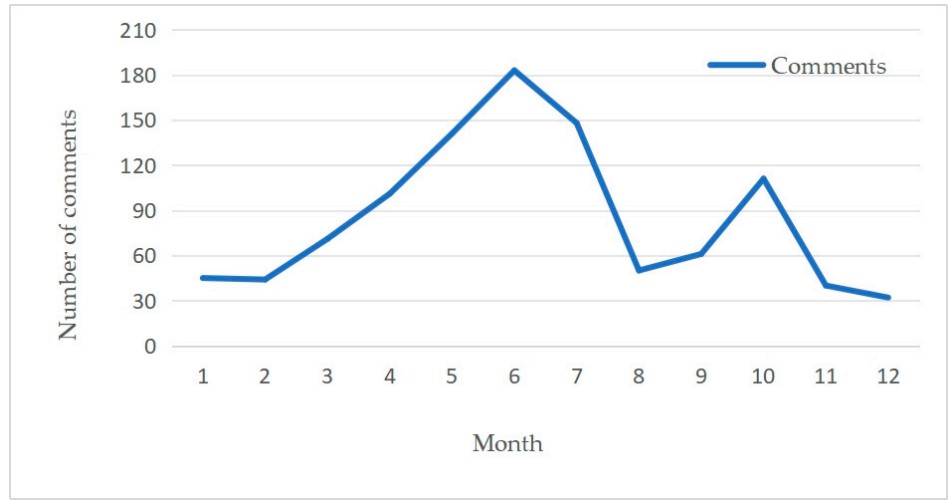

**Figure 6.** Monthly number of comments on Shijiazhuang Hutuo River Corridor (2022–2023).

The number of comments reached its peak in June. Figure 7 illustrates that user comments on the Hutuo River Corridor in Shijiazhuang are significantly higher in the summer compared to other seasons. It has been reported that on hot summer days, the sight

of water induces a refreshing and cool sensation [112,113], as expressed in the comment: "As soon as you get here, you can feel that the temperature is much lower than in the city". This experience attracts people to get as close to the river as possible [114]. Additionally, certain social activities draw people to the area. For instance, the music fountain opens during this time, becoming a highlight for enjoying the scenic Hutuo River Corridor on summer nights. "The evening lights are colorful and magnificent, a perfect combination of light, shadow, and music". Previous research indicates that summer visitors are more inclined to enjoy social relationships by the riverside, such as being accompanied by friends or family [74]. One individual described the Hutuo River Corridor as "A treasure camping ground perfect for gatherings with friends!" Therefore, social activities may contribute to the heightened vibrancy of the Hutuo River Corridor during the summer.

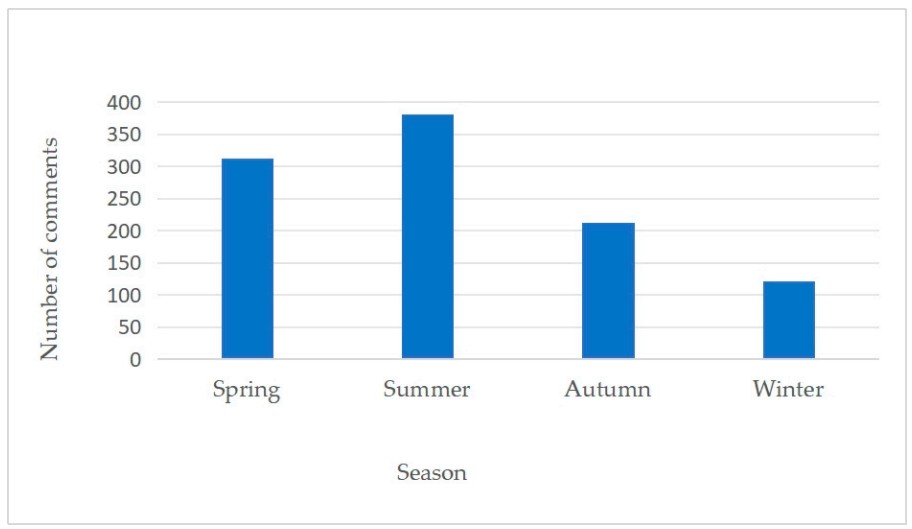

**Figure 7.** Seasonal number of comments on Shijiazhuang Hutuo River Corridor (2022–2023).

Subsequently, there was a sharp decline in the number of comments, reaching a nadir in August, primarily due to the impact of the flood season and high temperatures. During this period, Shijiazhuang experienced its rainy season, particularly from 27 July to 2 August 2023, when heavy rainfall resulted from the combined influence of cold and warm air and Typhoon Dujuan. This led to a significant rise in the Hutuo River's water level, necessitating the river corridor's closure. Naturally, this had a considerable impact on visitors' outings and activities. One comment described it: "Because of the flood season, the Hutuo River scenic area is blocked, many places are closed, and even under the bridges are sealed off. Water is released into the river. So, there are very few people and cars". Martín's research suggests that unfavorable weather conditions negatively affect thermal comfort, posing obstacles to outdoor recreational activities [115]. In Shijiazhuang, with the average temperature reaching 31–32 °C in August, users commonly expressed sentiments like "It is too hot" and "Over thirty degrees is indeed not suitable for going out".

In October, there is another minor peak in comments, driven by the blooming flower sea during Autumn, attracting citizens who come eagerly to take photos and check in. One person described it this month: "Following the scenic route of the flower sea, heading east past Taihang Street, you can see it. Now the silver grass and chrysanthemums are in full bloom"! While there may not be abundant descriptions of Autumn in the textual comments, they tend to be cheerful: "Autumn is a beautiful season, featuring a particularly eye-catching plant, the pink muhly grass. It is like dressing the earth in a magnificent red carpet. "The river water sparkles in the sunlight, shimmering with a golden glow; the leaves take on a touch of fiery red, gently swaying in the breeze as if nature has prepared a scroll of painting for us".

Research on winter leisure experiences indicates that seasonal environmental attributes and weather can influence people's outdoor leisure experiences [116]. Shijiazhuang, expe-

riencing distinct seasons, fully enters winter by December, with the grass turning yellow. Some describe this scene as "everything desolate, leaving only barrenness". This result aligns with findings by Cheng et al. [117] in the Beijing region. Haggarty et al. [118] attribute this to the shorter days and cold weather associated with winter. Additionally, studies show that children are less active in winter compared to other seasons, particularly in regions with cold and prolonged winters [119,120]. Parents often revolve their activities around their children, contributing to the reduced number of visitors during the winter. Nevertheless, some are still willing to come to the Hutuo River Corridor, "accompanying the elderly, taking children for a winter stroll, enjoying the winter sun;" "If the weather is good, take a walk by the river, or sit down for a cup of self-brought coffee, or brew a pot of hot tea, basking in the sun. It is quite pleasant".

(8) Weather

People's outdoor recreation is often affected by weather conditions. Visitors tend to prefer outings during weather characterized by "clear skies", "blue skies", "comfortable temperatures", "coolness", "neither too cold nor too hot", "warmth", "temperature rise", and "not too much sun". In early spring, people enjoy moments when the weather is "warm", "clear with a vast sky", "bright and beautiful", "with blue skies and white clouds", and "gentle breezes". It is a time for "soaking up the sun" and "breathing fresh air", and many express that "spring is the best season, not too cold or hot, suitable for barbecues". During summer and early autumn, individuals often complain about the "only drawback being excessive sun exposure, feeling like the sun is roasting them" and that "not many big trees, limited shade, and it is a bit too sunny in the summer". Consequently, they typically choose a day when the sun is not too intense, allowing for a full day of outdoor activities. However, some may complain about "more mosquitoes in the afternoon after rainfall and during cooler weather". In late autumn and winter, people tend to stay indoors with the arrival of cold air. Therefore, they only select days when the weather is "neither too cold nor too hot, suitable for sunbathing". Despite the management's organization of winter activities, which attracts many citizens, most still complain about "it being too cold" and prefer to stay outdoors only briefly. Nevertheless, some individuals maintain a positive attitude towards winter, appreciating the "icy Hutuo River in winter, which, without flowers, still has its charm", forming a "natural skiing ground". This optimistic perspective toward winter indicates that the Hutuo River attracts people seeking natural enjoyment even in the cold season.

## 4. Discussion

### 4.1. Factors Influencing Recreational Experiences in Urban River Corridors

This study delves into the diverse perspectives of urban river corridor users regarding the social and physical environmental factors influencing recreational experiences, leveraging data obtained from online reviews. It reveals that urban river corridors attract users for various purposes at different times, particularly during summer evenings, winter nights, and tranquil mornings, appealing to numerous tourists and residents alike. This phenomenon is intertwined with China's work schedule constraints, as individuals often prefer nearby free green spaces for outdoor leisure activities during weekends or holidays. This observation finds affirmation in three similar studies conducted in China [121–123]. However, heightened visitor numbers during holidays and weekends underscore critical issues related to parking, as emphasized in user comments. The availability of leisure time significantly influences the duration of recreational activities along urban river corridors [124], with prolonged stays often reflecting the allure of the landscape [69].

Various physical environmental factors influence users' activities in urban river corridors, including natural surroundings, infrastructure, and ambiance [125]. The primary qualities or benefits that attract individuals to green spaces or enable them to engage in preferred activities are considered contributors to the biodiversity of natural environments and landscape aesthetics [126], thereby enhancing aesthetic quality [127,128]. However, unclean, disorderly, and substandard environments directly impact people's physical

health and contribute to feelings of insecurity [67], potentially diminishing user satisfaction. Therefore, maintaining cleanliness, orderliness, and high environmental quality along river corridors is crucial for enhancing user experiences. Additionally, increasing tree canopy cover provides biodiversity and creates a tranquil environment, attracting individuals and positively influencing their interactions with nature [72,129]. However, earlier studies suggest that dense vegetation may induce fear of crime [130,131], highlighting the need to balance perceptions of safety and natural ambiance in the design of river corridors. This study also yields conclusions not previously addressed in the literature. The lack of shade, mosquito disturbances in summer, and climatic conditions on winter nights significantly impact user experiences.

User engagement with urban river corridors is influenced by accessibility, including the continuity of pathways and the availability of infrastructure [132]. Research indicates that the accessibility of urban waterfront spaces is crucial for enhancing user experiences. Due to China's unique social system, the accessibility of urban waterfront spaces differs from that in Western countries. For instance, the accessibility of urban river corridors in China may not be influenced by land privatization, thus rendering the services and opportunities in waterfront spaces perceived as equitable. Numerous riverside development initiatives emphasize the necessity of constructing uninterrupted pathways along water bodies to significantly enhance accessibility [133,134]. Subramanian and Jana's study [135] confirms that universal design elements, such as paved walkways, universal entrances, handrails, ramps, and tactile flooring, substantially enhance the perceived attractiveness of leisure open space pathways. The availability of such infrastructure also crucially impacts the accessibility of river corridors. Furthermore, in waterfront public areas, ample and aesthetically pleasing wayfinding and signage facilities are essential for enhancing accessibility, a point corroborated by Barnett et al.'s research [136]. Additionally, gaining in-depth insights into the cultural background and geographical significance of landmark structures enhances visitors' cultural experiences during exploration. Therefore, urban planning and infrastructure maintenance should prioritize continuity, minimize disruptions, and provide ample and aesthetically pleasing wayfinding and signage facilities to ensure users can access urban river corridors conveniently and smoothly.

### 4.2. Insights into Urban River Corridor Management

Despite visitors generally holding favorable opinions of urban river corridors, there remains room for improvement. To address this situation, a series of measures should be implemented to enhance user experience and environmental quality. Firstly, there is a need to intensify management and maintenance efforts concerning vegetation, restrooms, accessibility facilities, and recreational amenities to ensure the upkeep of the corridor's environmental quality. Secondly, managerial service levels, including fee reductions and strengthened sanitation management, should be elevated to enhance service quality and user satisfaction. Additionally, improving transportation convenience is imperative, necessitating the optimization of traffic management, the augmentation of parking spaces, and the enhancement of parking management mechanisms, such as the introduction of intelligent parking systems or the implementation of differential pricing policies [137,138]. Simultaneously, to bolster the connectivity of urban river corridors, optimizing public transportation routes and installing bike lanes are necessary measures to enhance user mobility. Through the collaborative efforts of government departments, urban planners, community organizations, and the public, a more pleasant, convenient, and sustainable environment for urban river corridors can be created, providing residents and tourists with enhanced leisure experiences [139,140].

### 4.3. Strengths and Limitations

Traditionally, the influence of environmental factors on users of different types of urban green spaces has been studied based on users' socio-demographic characteristics, often through surveys or interviews. However, this study takes a novel approach, inte-

grating the social-ecological model with qualitative analysis based on social media data, to comprehensively investigate the social and physical factors affecting leisure experiences along urban river corridors. This innovative method has proven to be meaningful [141,142], offering a fresh perspective on the subject. By categorizing and analyzing social media data thematically, this study delves into individuals' subjective experiences and perceptions. Utilizing extensive text data mining techniques, the research not only rapidly captures environmental conditions and recreational experience demands but also helps bridge the gap in public participation in landscape design processes. Ultimately, the study aims to facilitate the provision of higher-quality urban river corridor services that meet citizen expectations, promote urban sustainable development, and enhance the overall well-being of urban residents.

While this study provides valuable insights, it also highlights the need for further research. Firstly, the study primarily focuses on Shijiazhuang, an inland Chinese city characterized by distinct seasons and limited water resources, raising concerns about the generalizability of the findings to other cities. Although some common attributes may apply across cultures based on evolutionary foundations, further research is necessary in different regions. Secondly, social media users are predominantly composed of young and middle-aged individuals and may not fully represent all demographic groups, excluding certain populations who do not use applications. According to statistics from 2022 on the age structure of Chinese internet users, individuals aged 60 and above account for only 11.3% [143], indicating that the majority of older adults do not use the internet. Consequently, relying solely on social media data may result in sample bias, capturing only a portion of the population's viewpoints and feedback. In future research and analysis, it is essential to carefully consider this limitation and integrate other data sources to ensure a more comprehensive and accurate understanding of different age groups, social backgrounds, and usage habits. Additionally, due to the diversity of big data, the data sources used in our study are limited. Therefore, future research should incorporate multiple data sources to assess and improve the accuracy of the results. Your potential contribution to this field is crucial in addressing these limitations and advancing our understanding of urban river corridors.

## 5. Conclusions

This study employs data mining techniques to extract high-frequency keywords and conduct topic and content analysis on online reviews sourced from Dianping. The primary objective is to elucidate visitor perspectives and attitudes by tapping into the abundant information in these comments. This study sheds light on visitor preferences, constraints, and the transformative impact of ecological restoration projects on urban river corridors. Firstly, while spring and summer attract more visitors, winter holds an appreciable attitude despite less visitation. Visitors prefer weekends, evenings, and mornings, and the visiting time is closely linked to weather conditions. Secondly, the Hutuo River Corridor receives widespread acclaim among visitors, who express positive views on its natural environment, planning and design, convenient facilities, and transportation conditions. Notably, the successful implementation of the Hutuo River ecological restoration project has transformed the area into an ideal place for citizens' leisure and fitness activities. Thirdly, accessibility is a crucial factor influencing visitor usage of the Hutuo River Corridor. Visitors generally perceive transportation and parking convenience positively, although insufficient parking resources may arise during peak seasons. Improving the accessibility of public transportation, especially during peak hours, can facilitate more citizens' reaching and enjoying the area conveniently. Additionally, constraint factors are mentioned in user comments, such as staircase width, limited space at the tower top, and various weather and seasonal considerations.

In conclusion, analyzing social media data offers profound insights into optimizing the urban river corridor. Beyond the specific case, the findings suggest broader implications for urban recreational spaces. It contributes to understanding visitor behavior, preferences, and the importance of accessibility in enhancing urban recreational spaces' usability. However,

several questions still need to be answered, such as the long-term sustainability of ecological restoration projects and the potential impacts of climate change on visitor patterns. Future research could explore these aspects further. Furthermore, this study underscores the significance of integrating visitor feedback into urban planning and management processes to enhance public space accessibility and visitor experiences. Recommendations for blue–green infrastructure practices, management strategies, and sustainable development initiatives can be drawn from these insights, contributing to advancing the field of urban blue–green space management and design.

**Supplementary Materials:** The following supporting information can be downloaded at: https://www. mdpi.com/article/10.3390/app14104086/s1, Table S1. High-frequency feature words in user reviews of the Hutuo River Corridor in Shijiazhuang (urban section): top 100 rankings. Table S2. Frequency list of LDA feature words for each theme (top 30). Table S3. LDA theme description and comment examples.

**Author Contributions:** Conceptualization and methodology, L.S., S.M. and C.D.; writing—original draft preparation, L.S.; formal analysis, investigation, and data curation, L.S.; validation, S.M., M.J.M.Y. and C.D.; writing—review and editing, S.M., M.J.M.Y. and C.D.; supervision, S.M. and M.J.M.Y. All authors have read and agreed to the published version of the manuscript.

**Funding:** This research received no external funding.

**Institutional Review Board Statement:** The study was conducted in accordance with the Declaration of Helsinki, and approved by the Ethics Committee for Research involving Human Subjects of University Putra Malaysia (protocol code JKEUPM-2023-1338).

**Informed Consent Statement:** Informed consent was obtained from all subjects involved in the study.

**Data Availability Statement:** The original contributions presented in the study are included in the article/Supplementary Material, further inquiries can be directed to the corresponding author.

**Acknowledgments:** We would like to thank all the literature and previous research cited in this study. Their work has provided a solid foundation for our review, enabling us to delve deeper into the issues relevant to this field.

**Conflicts of Interest:** The authors declare no conflicts of interest.

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
