# Peer review of "Exploring Factors Influencing Recreational Experiences of Urban River Corridors Based on Social Media Data"

_applsci, doi:10.3390/app14104086_

Round 1

Reviewer 1 Report

Comments and Suggestions for Authors

Thank you very much for the interesting read and the presented work. However, a number of points need to be addressed and worked on and some points need attention before the manuscript can be considered for publication. As presented, it is very descriptive and for several aspects, it would be worthwhile to include some more substantial work steps and more analyses as well as reflecting theoretical foundations to embed the work in a broader context to improve and make the work much more relevant for an international audience.

The most serious issue is that the work contains data collected from humans. Usually, such work requires special attention to be in line with common ethical standards e.g. the World Medical Association Declaration of Helsinki. How is privacy maintained during the research process e.g. disconnecting personal information from the collected Dianping entries? Did participants give their active consent to make use of their data? There is no hint in the paper. It would be needed in a form of an ethical statement declaring that the authors of the study have followed these guidelines or principle according to international and national standards and regulations and/or an approval by an ethical committee to follow good practice working with such materials. There is no explanation or description how privacy was maintained and disconnecting private data from the research objects.

Data basis: When using Dianping, it would be interesting to describe what section of the population you capture. How many persons use it and what are the rough demography of this local life information and rating platform. It is very briefly mentioned at the end of the manuscript but remains very vague. A critical reflection of using this source and what is achieved as well as potential biases would be useful in a proper discussion section where one part would be reflecting about the methodology and collected data in more depth and also compare it with findings in literature where similar or comparable media sources were used or multiple approaches have been conducted.

When analyzing around 3000 data sets with reviews, it would be interesting to create more quantitative outputs of what was described. How many times different features were mentioned? Could it be linked to spatial features? The gained information then could be transferred and analyzed in a spatial context to create interesting valuable spatial information e.g. providing heat maps where there it is possible to show where there are spaces and areas inside the park that are more frequently mentioned for certain features. This would be especially interesting for aspects of scenery or recreational activities. The work as presented is very descriptive and does not dive deeper and draw analyses here. A lot of potential the data offers for interpretation and analyses is not unveiled in presenting the material only in this descritive way.

The work largely is describing the case study and outcomes of the text analyses in an exhaustive way. Several references refer to previous work conducted. A look beyond the own case and presenting could be better developed. As presented, there is only a reference here and there to other cases in the combined result and discussion section.

A sound theoretical framework is missing, for example embedding the work in theories such as landscape perception and preferences. As presented, it is nice for a project report but to put the work in a broader context and for a scientific journal with an international audience, presenting the work needs to be embedded in theoretical frameworks and the current discussions to make it interesting for a broader audience. Doing this by further elaborating the presented work could contribute nicely here and would increase the relevance for a broader scientific discussion.

This implies to elaborate the introduction in a better way and to better explain, what are research and knowledge gaps, what is new and what would be the contribution of this work? In the text, a bit is somewhat described. But it needs to be better elaborated and highlighted how your work differs from the other works that have made use of social media and what would be the added value of the presented study.

A separate discussion section including literature is missing. Besides reflecting on the methodologies, how does the work contribute and embed in the scientific discussions about qualities of landscapes and restored places and draw comparisons with other cases? There are quite some projects both in China and around the globe with river corridor restoration measures or creation of blue green or green corridors. Besides referring to other cases here and there in the combined results and discussion section, it would be good to have a proper discussion section and a part in there dedicated to summarize and have a look from an overall perspective.

Finally, for the conclusion section as presented, it largely reflects and circles only around the case study. To show the relevance beyond the study area, it should be concluded what can be learned from the case in a broader context. To which research gaps and scientific discussions the presented work can contribute? Which questions were answered and which remain open? Where would be research needed in the future? Depending on the messages that the authors want to give, the final section of the paper could also give provide recommendations and implication for practice and the management and development of blue-green infrastructure.

To conclude, the material and study has a lot of potential but I´d reccomend to conduct additional analyses and work and to better elaborate several sections to improve the manuscript to unveil the full potentials of the collected materials and to make it really meaningful for an international readership. This work would require more than some additional controls or revisions and investing more time than a few weeks to revise which are usually given by the journal.

My reccomendation is therefore rejecting the manuscript in the current form. Firstmost, authors need to create clarity if they followed overall general ethical standards e.g. described such as the World Medical Association Declaration of Helsinki which can give a good guideline for such work. This declaration is presenting ethical principles for medical research involving human subjects and in many ways what is described there is also valid for working with collecting overall data from humans such as data from social media. Also other ethical guidelines and country specific legal regulations would be of relevance and need to be considered and followed and referred to as such in a seperate ethical statement section. Please make sure if you have followed such principles in your work and also describe it in the methodology how you ensured these principles.

If this aspect can be sufficiently clarified, I would like to strongly encourage the authors to take some more time to work on the manuscript, add additional analyses and then resubmit - these additional efforts would be really worth it both in terms of theoretical embedding and looking at the potentials of the data you have collected.

Reviewer 2 Report

Comments and Suggestions for Authors

This article measures the sentiment of local population towards Hutuo River Corridor in China.  This is an interesting topic, using social media data mining to determine usage and opinions of an urban green space.   The article is well situated in the academic literature, demonstrating how their research fills gaps in the academic discourse.  Their methods are quite innovative and appropriate for their research question.  I personally see the methods are the most significant contribution of the article, as it provides an approach to social media mining that is useful in various studies of public sentiment.  Their results are well situated in the academic discourse, providing examples of how each theme is tied to the academic literature on green space usage.  

Overall, I found the article to be well researched and well written.  The topic is timely and innovative.  I have no significant suggestions for the article.  Thank you for the opportunity to review the article.  Best of luck in your future research.  

Reviewer 3 Report

Comments and Suggestions for Authors

General comments 

This study investigates the Hutuo River Corridor in Shijiazhuang, China by collecting valid reviews from Dianping, a prominent review platform. The authors provide a text- 15 based thematic model and conducted content analysis using this dataset. In my view, the topic has the originality to be considered for publication as it covers the gap in the literature.

The authors provided a comprehensive literature review although some articles should be added.

The findings indicate that the feasibility of employing social media data to study visitors' recreational experiences along urban river corridors. The conclusion has written in a proper format.

Some missing references should be added (please see my comments above).

Tables and figures are legible.

Specific comments

The manuscript entitled “Exploring Factors Influencing Recreational Experiences of Urban River Corridors: A Qualitative Ecological Perspective” seems to be acceptable to be considered for publication in Sustainability, but it would be better if some issues be regarded in the revision.
1. Since this paper is not a literature review, in lieu of referring to a vast number of previous works, which have been surveyed by other researchers, mention the latest related works.

A Biobjective Model for Integrated Inventory and Transportation at Tactical and Operational Levels With Green Constraints. IEEE Transactions on Engineering Management.

 Assessing the cultural ecosystem services value of protected areas considering stakeholders’ preferences and trade-offs—taking the Xin’an River Landscape Corridor Scenic Area as an example. International Journal of Environmental Research and Public Health, 19(21), p.13968.

Direct and indirect emissions: a bi-objective model for hybrid vehicle routing problem. Journal of Business Economics, 94(3), pp.413-436.

The Wild and Scenic Rivers Act at 50: Managers’ views of actions, barriers and partnerships. Journal of Outdoor Recreation and Tourism, 37, p.100459.

2. The transitions from topic to topic in the paper seem to be a little sudden. In other words, while reading about a topic, the text suddenly starts to mention something quite different. It is suggested to smooth these transitions from topic to topic where possible.
3. Please indicate the contributions in more detail, specifically in comparison with the latest research papers.
4. Above all, please polish the title to be in line with the contents of this manuscript.

Comments on the Quality of English Language

 Moderate editing of English language required

Reviewer 4 Report

Comments and Suggestions for Authors

This paper explores the social and physical factors that promote or hinder recreational experiences along an urban river corridor in China, using extensive text data mining and content analysis.

The Introduction section is very well structured. However, there are some issues which need to be addressed. What is the research gap that this study fills? The problem statement should be emphasized.

The methods are well described, as well as the study site and research procedure.

The Results and Discussion section is well structured and the information is illustrated accordingly. However, the theoretical and managerial implications need to be discussed.

Round 2

Reviewer 1 Report

Comments and Suggestions for Authors

Dear Authors,

most of the comments have been further elaborated and the text improved a lot. Looking at the revised text, some quantitative analyses would be nice to be added but would require additional time, work and need for more investigation and understand that you do not further elaborate. The main remaining point is related to an ethical statement as you work with data of humans and a section in the manuscript with the an Ethical Statement is still missing. I suggest to cross check other papers (e.g. from medicine) how such a section is usually elaborated. I see the approaval as supplementary material but you do not need to provide this. It is about mainly adding a standardized section/chapter called "Ethics" or "Ethical Statement" to your paper with some reflection on how you followed ethical guidelines, ensured to have consent of the persons providing the data, even if it is mutual consent. E.g. by agreeing for such uses when registering to the social media platform. Also, it needs to be explained and how was privacy maintained and which legal basis is relevant. It does not need to be very long and just a shorter section. As said, cross-check with other work how such a section is elaborated and what is usually in it.
